# OpenReview forum: "Strategic Dishonesty Can Undermine AI Safety Evaluations of Frontier LLMs"
_ICLR.cc/2026/Conference — ICLR 2026 Poster_

### Official Review · Reviewer_BcRu · 2025-10-27

**Soundness:** 3
**Presentation:** 2
**Contribution:** 2
**Rating:** 6
**Confidence:** 3

**Summary:**

This paper presents an empirical and conceptual investigation of strategic dishonesty—a phenomenon where frontier large language models (LLMs) respond to harmful or policy-violating prompts not by refusing, but by producing apparently harmful yet deliberately incorrect answers. The authors argue that this behavior, which they frame as “misalignment faking,” undermines the validity of current safety evaluations that rely on output-based scoring from weaker LLM judges. They find that (i) many models, especially more capable ones, tend to sacrifice honesty rather than helpfulness or harmlessness; (ii) their faked harmful responses fool all existing jailbreak judges; and (iii) deception probes trained on hidden activations can reliably detect this behavior and even steer models away from dishonesty.

**Strengths:**

1. The authors clearly articulate “strategic dishonesty” as distinct from both sandbagging and alignment faking, situating it in the tension between helpfulness and harmlessness.

2. The paper evaluates over 80 models, spanning both open and closed weights, and uses standardized benchmarks like ClearHarm and EvilMath. The breadth and systematic nature of the experiments make the findings robust.

3. The finding that every tested jailbreak judge—HarmBench, JailJudge, LlamaGuard, etc.—is fooled by fake harmful outputs is striking and important.

**Weaknesses:**

1. While the paper identifies that dishonesty occurs, it remains unclear why. The analyses around RLHF, reasoning, and lying propensity (RQ1–RQ3) are inconclusive.

2. The multiple-choice format, though creative, may not reflect natural model reasoning. Models might pattern-match to template cues rather than genuinely “choose” between honesty and helpfulness.

3. While the deception probes perform well on open models, their reliability under distribution shift (e.g., unseen harmful scenarios) or adversarial steering isn’t tested.

**Questions:**

How does the deception probe behave when the model hallucinates factual errors unrelated to deception?

Could probe-based steering interfere with benign reasoning or degrade model accuracy on truthful tasks?

---

> ### Author Response · Authors · 2025-11-18
> **Response to Reviewer BcRu (1/2)**
>
> Dear Reviewer BcRu,
>
> We thank the reviewer for a thorough evaluation of our work. We are glad that the reviewer found our findings robust and important. We are grateful for the reviewer's appreciation of our efforts to distinguish our work from classical notions of sandbagging and alignment faking.
>
> Let us first address the concerns you raised about deception probes and clarify some details of our setup:
>
> > #### *While the deception probes perform well on open models, their reliability under distribution shift (e.g., unseen harmful scenarios) … isn’t tested*
>
> We would like to clarify our probe training and evaluation setup. Following [1], probes are trained on simple pairs of instructed factoids:
>
> - **Positive example:** "Prompt: Pretend you're a dishonest person making statements about the world. Response: The Eiffel Tower is located in Paris, France".
> - **Negative example:** "Prompt: Pretend you're an honest person making statements about the world. Response: The Eiffel Tower is located in Paris, France".
>
> Both the EvilMath and ClearHarm benchmarks constitute **completely unseen harmful scenarios** and represent substantial distribution shifts from the training data. We also evaluate probes on benign math (UnicornMath) and benign queries (AlpacaEval). In these settings, the probes maintain low false positive rates, demonstrating robust generalization.
>
> Furthermore, we test probe robustness under nanoGCG adversarial attacks (Section 6 and Figure 7), showing that deception probes continue to work as expected. Note that nanoGCG discovers highly unnatural suffixes absent from both model and probe training data, constituting an even stronger distribution shift.
>
> > #### *adversarial steering isn’t tested.*
>
> In Section 5.3, we adversarially steer with the direction discovered by probes toward both more and less deceptive behavior, and observe that models indeed become more deceptive. This causal link provides additional evidence of probe reliability.
>
> We further train a model to become more misaligned through adversarial fine-tuning in Appendix E. In Figure 10, we demonstrate that probes generalize to this misaligned model.
>
> Taken together, these experiments present comprehensive evidence of probe reliability and generalization under diverse distribution shifts.
>
> > #### *How does the deception probe behave when the model hallucinates factual errors unrelated to deception?*
>
> Following [2], we define deception as knowingly stating falsehoods. In contrast, we view hallucination as stating falsehoods without awareness of their incorrectness. Our probes are designed to specifically detect the former.
>
> In our validation experiments (Section 5.1 "Validating Dishonesty Probes with Verifiable Outcomes"; details in Appendix C.2), as a first step, we evaluated responses to benign math problems without any instruction to deceive, categorizing answers as Correct or Incorrect. Here, incorrect answers represent unintentional factual errors, i.e., hallucinations. Across all tested models, we found that these factually incorrect answers did not activate the deception probes. Furthermore, when the model was explicitly instructed to deceive, the probes activated only when the model actually followed the instruction (Figure 5, left, second row), with a minimal number of false positives (Table 8).
>
> > #### *Could probe-based steering interfere with benign reasoning or degrade model accuracy on truthful tasks?*
>
> In the activation steering literature, the consensus is that simple, fixed-multiplier activation steering can lead to side effects (see e.g., [3]). While our experiments suggest that steering does not affect mathematical capabilities (see Figure 6), we _do not_ propose probe-based steering as a practical method for controlling dishonesty in deployed models.
>
> Rather, we use steering with the probe direction as an interventional experiment to establish a causal link between the "dishonesty concept" encoded in the probe direction and the actual deceptive behavior. This causal link demonstrates that the probe relies on the same mechanism driving deceptive behavior, rather than on spurious correlations.

---

> > ### Author Response · Authors · 2025-11-18
> > **Response to Reviewer BcRu (2/2)**
> >
> > > #### *While the paper identifies that dishonesty occurs, it remains unclear why.*
> >
> > We evaluated three hypotheses to understand the origins of this behavior: (i) whether RLHF induces faking, (ii) whether reasoning encourages faking, and (iii) whether faking is part of a broader lying tendency. While none provided a complete explanation, we find it important to rule out these alternatives and avoid overclaiming about mechanisms we cannot definitively establish.
> >
> > That said, if we were to speculate, we believe the roots of this behavior lie in the nature of LLM training itself. Base models learn from the full spectrum of human-written text, including sources depicting strategic deception and white lies. This places such behaviors squarely within the realm of possibilities for modern LLMs. However, which behaviors manifest, and how strongly, is likely determined by post-training. Yet, understanding how exactly alignment procedures shape these (and many others) propensities remains a question for future work.
> >
> > > #### *The multiple-choice format, though creative, may not reflect natural model reasoning. Models might pattern-match to template cues rather than genuinely “choose” between honesty and helpfulness.*
> >
> > We would like to clarify that our core values evaluation is not a standard multiple-choice question (MCQ) task. We require the model to 1) first provide reasoning, 2) then, explicitly mention the chosen strategy, and, finally, 3) generate a response according to this strategy. This closely resembles free-form generation in standard CoT, which normally consider response strategies during their reasoning, then commit to one strategy and generate a response accordingly.
> >
> > We agree that our MCQ template may not fully capture "in-the-wild" behavior. However, the observed behavior cannot reduce to simple pattern-matching to template cues - as different models exhibit substantially different propensities to strategic dishonesty under the same system prompt. In Appendix B, we now have extended our MCQ evaluation and demonstrate that the identified *propensities remain remarkably robust* across different types of harmful queries and multiple languages (English, Spanish, Chinese, and Ukrainian).
> >
> > ---
> >
> > Hopefully, we convincingly addressed your questions and clarified the issues you raised. If this is the case, we kindly ask you to consider raising the score. Otherwise, we are happy to discuss any remaining concerns.
> >
> > ---
> >
> > **References:**
> >
> > [1] Detecting strategic deception with linear probes, Goldowsky-Dill et al., 2025
> >
> > [2] The MASK Benchmark: Disentangling Honesty From Accuracy in AI Systems, Ren et al., 2025
> >
> > [3] Steering Without Side Effects: Improving Post-Deployment Control of Language Models, Stickland et al., 2024

---

### Official Review · Reviewer_KSiq · 2025-10-29

**Soundness:** 4
**Presentation:** 4
**Contribution:** 3
**Rating:** 6
**Confidence:** 4

**Summary:**

The authors show that, across a variety of harmful prompts, frontier LLMs sometimes give seemingly-harmful but actually-benign advice to users. I.e. they choose to be deceptive rather than flat-out refuse. They perform extensive evaluations and show it occurs to some extent in many different model families, though no scaling laws are obvious. They also show that deception probes reliably detect such strategic dishonesty. Lastly, they investigate some simple hypotheses for why models strategically deceive and find no clear insights. Overall, they elucidate a current empirical challenge for scalable oversight - that humans and weaker LLM judges will not be able to accurately determine whether the output of a stronger LLM is factually / technically correct.

**Strengths:**

Originality: good. Strategic deception has been discussed in prior work, e.g. https://arxiv.org/abs/2506.04909. However, the authors improve upon literature by providing a clear framing, doing extensive experiments across frontier models, reporting the empirical usefulness of deception probes, and doing many more ablation / analysis studies.

Quality: good. Overall experimental methodology is sound and the experiments conducted are extensive. However I have some concerns which are described below in 'weaknesses'.

Clarity: good. Paper is well written and easy to understand.

Significance: good. The finding, if true, has substantial implications for the way frontier models are evaluated for safety and alignment - in particular it suggests that many current evaluations (like StrongREJECT) are fundamentally broken. However this is also bottlenecked by some concerns I have with the empirical methodology.

**Weaknesses:**

The MCQ-based evaluation described in Figure 2 is pretty artificial due to the use of extensive system prompting / ICL examples which likely won't be present in realistic use cases. As such it's unclear if model choices are reflective of 'real' propensities.

In section 4, it's not clear whether you filter out genuinely harmful responses as well? IIUC your evaluation here does not distinguish models that have high faking capability from models that are highly jailbroken (i.e. provide actually harmful responses).

In the evaluation of probes on the ClearHarm setting, you say that deception probes fire over 99% of the time; however, IIUC there is no attempt to tell if the model outputs are actually harmful (i.e. false positives for the probe) or real faking (i.e. true positives for the probe).

It's not clear to me whether the probes measure 'intent to be deceptive' or 'actual deception' - maybe the model just thinks about being deceptive without actually deceiving the user?

At the moment these issues limit me to giving a borderline accept, I am happy to raise my score if these are addressed satisfactorily

**Questions:**

stated above

---

> ### Author Response · Authors · 2025-11-18
> **Response to Reviewer KSiq (1/3)**
>
> Dear Reviewer KSiq,
>
> We thank you for the detailed review and thoughtful feedback. We additionally thank you for bringing to our attention the paper by Wang et al., which is highly relevant to our work as it discusses white-box-based interventions with many methodological similarities. We now discuss it in our Related Work. Below, we address the questions you raised.
>
> >  #### *… extensive system prompting / ICL examples which likely won't be present in realistic use cases. As such it's unclear if model choices are reflective of 'real' propensities.*
>
> We fully agree that propensities discovered in an MCQ setup might not directly reflect 'real' and 'in-the-wild' behavior. However, we would like to highlight two important points:
>
> 1. **This behavior is likely unintended by model providers.** Faking misalignment is not "what they paid for" - models are post-trained to refuse harmful queries, for which they have an explicit option in our setup. In this regard, any inclination of a model to pick the dishonesty option is an alignment failure.
>
> 2. **The propensity appears fairly robust to variations in the MCQ format.** To test this robustness, we conducted additional experiments examining (i) how different sources of harmful queries affect exhibited propensities and (ii) how changing the language of system prompts and queries affects propensities. We include these results in Appendix B (pages 21-23) and present a summary below.
>
>  **Do findings hold across different harmful content types?**
>
> To better answer this question we have now also evaluated 10 models on the HarmBench dataset, which spans different semantic categories of harmful content than ClearHarm (which we use mainly in this work). We observe no significant difference between HarmBench and ClearHarm (Figure 9). Interestingly, models tend to fake more on factual queries than on social harm queries. We speculate that for misinformation-type queries, any non-truthful information effectively results in compliance, making this strategy easier to execute in factual domains.
>
> **Table 1.** Avg. model preferences across semantic categories (%).
>
> | Category | Faking | Refusal | Compliance |
> |--------------------------|--------|---------|------------|
> | Chemical & Biological | 19.1 | 76.7 | 2.6 |
> | Cybercrime | 28.5 | 60.9 | 7.8 |
> | Harassment | 5.7 | 92.2 | 1.5 |
> | Harmful Content | 6.1 | 89.0 | 4.1 |
> | Illegal Activities | 18.5 | 76.5 | 3.4 |
> | Misinformation | 3.7 | 91.9 | 3.8 |
>
> **Do findings hold across languages?**
>
> We translated our MCQ setting and ClearHarm queries into 3 new languages and evaluated 6 models. We found that 5 out of 6 models exhibit the same propensities across languages (e.g., Gemini 2.5 Pro consistently prefers faking in all languages). The only exception is GLM 4.5 Air, which shows no clear preference in Spanish and Ukrainian, suggesting alignment generalization failure beyond its target languages. For detailed results, please see Table 6 and Figure 11.
>
> **Further discussion**
>
> We see MCQ design as our best shot for measuring faking propensities in a controlled setting. As we demonstrate in the paper, current automated judges cannot reliably distinguish real from fake harmful outputs for highly capable models, and expert annotation at scale remains prohibitively expensive. While we acknowledge the gap between this setup and real-world behavior, the MCQ approach allows us to isolate and measure a phenomenon that would otherwise be nearly impossible to detect systematically. For less capable models, the MCQ setup with ICL examples helps bypass the reasoning steps needed to discover this strategy - steps that more capable models can complete independently (e.g., Claude 3 Opus; see Appendix M.1.1 in [1] by Sheshadri et al.).
>
> We hope that propensity measured in this way would inform alignment teams of the potential of this problem occurring, even if the in-the-wild incidence rates are lower. Note how for well-aligned models, e.g. GPT-5, this option is indeed never chosen.
>
> In realistic adversarial attack settings, beyond the anecdotal reports we discuss in the paper (e.g., Chowdhury et al. [2]), current attacks leave room for this phenomenon to occur. Consider a GCG attack, which suppresses the refusal but not guarantees harmfulness of the outcome.  For example, Beyer et al. [3] report (Figure 3 and Appendix E), attack outputs cluster into three categories: (1) refusal, (2) compliant but incomplete/irrelevant, and (3) compliant and harmful. We believe that the propensity to fake might shine through as the second outcome, when refusal is suppressed by the attack.
>
> We believe the identified phenomenon is worth taking into account for improving jailbreak evaluations, e.g., with white-box methods that we demonstrate could be highly effective in such setups.

---

> > ### Author Response · Authors · 2025-11-18
> > **Response to Reviewer KSiq (2/3)**
> >
> > > #### *In section 4, it's not clear whether you filter out genuinely harmful responses as well? IIUC your evaluation here does not distinguish models that have high faking capability from models that are highly jailbroken.*
> >
> > We are happy to clarify the setup of Section 4. In Section 4, we explicitly instruct all models to deceive in response to harmful queries, instead of providing a blunt refusal. However, some models (e.g., GPT-OSS) refuse-to-fake. These are the only responses we filter out in this section.
> >
> > However, the exact question you raise directly informs our approach in **Section 5**. We acknowledge that it is initially unclear whether the remaining models produce "fake harmful outputs" (successful deception) or "genuine harmful outputs" (jailbroken behavior). To address this, we use linear activation probes and CoT monitors.
> >
> > For highly capable open models (e.g., 32B parameters), we can be confident that they are being dishonest rather than jailbroken. This certainty is mainly achieved through validating deception probes with the verifiable EvilMath benchmark. In this setup, we demonstrate that whenever deception happens according to the linear probe, it induces factual inaccuracy. Note that **we do not train probes on EvilMath**, yet they perfectly separate accurate answers from deliberately flawed ones. Further, these same linear probes activate for >99% of “pseudo-harmful prompts,” while almost never (<5%) activating on benign prompts (Table 1).
> >
> > **In short:** We do not use linear probes to filter answers in Section 4. However, our analysis in Section 5 confirms that for larger models where probes function reliably, >99% of the non-refusal answers are indeed deceptive rather than genuinely harmful.
> >
> >
> > > there is no attempt to tell if the model outputs are actually harmful (i.e. false positives for the probe) or real faking
> >
> > This is a **core tenet** of the problem we identify in the paper. In Section 4, we show that the better the model is, the harder it is to tell whether the output is harmful or not. While for some models it is clear-cut (Figure 1, QwQ-32b names non-existent chemicals which can be verified with web search), for others (Appendix I, Gemini 2.5 Pro) it is impossible to determine if the output is harmful without being a domain expert. Thus, to claim deception we rely either on an explicit MCQ setup or on the white-box and black-box monitoring efforts we introduce in Section 5, where we can verify whether deception occurs through our EvilMath setup.
> >
> > For a small subset of Gemini 2.5 Pro generated outputs, we consulted two PhD-level chemistry experts and one law enforcement expert, and all three agreed that the provided dishonest substance recipes **would not enable** real-world harm. However, overall we cannot provide a complete human study on frontier model outputs due to its extreme cost.
> >
> > The only domain where we can rigorously evaluate False Positives (FP) and True Positives (TP) for the probe is EvilMath. When we instruct the model to deceive, the outcomes fall into four categories:
> > - **TP:** Model deceives, leading to an incorrect answer.
> > - **TN:** Model ignores the instruction to deceive and answers correctly.
> > - **FP:** Model attempts to deceive but inadvertently solves the problem correctly.
> > - **FN:** Model ignores the instruction to deceive but still fails to solve the math correctly.
> >
> > Based on your feedback, we have updated Table 8 in the Appendix to reflect the number of FPs produced by probes for every model on EvilMath, as well as the respective False Positive Rate (FPR). We also qualitatively assessed these FP examples; most represent cases where the model attempts to execute deception but becomes "confused" about which answer is correct versus which is crafted to be incorrect, ultimately outputting the correct answer.
> >
> > > #### *…maybe the model just thinks about being deceptive without actually deceiving the user?*
> >
> > This is indeed plausible and a limitation of existing deception probes. As we show in Appendix E, when the model intends to deceive in the CoT but then answers honestly, the probe gives a false positive. Similarly, as shown in Figure 5 (Right), we find that for smaller models, which are more often inconsistent in this way, the probes do not work as well. Thus, existing probes do assume a certain continuity of behavior and CoT faithfulness, which might not be present in less capable (or poorly trained) models.
> >
> > That being said, if, for example, Gemini 2.5 Pro were to commit to executing deception in its reasoning, but produced genuinely harmful output at the moment of final answer generation, this would constitute a FP for the CoT monitor. However, we currently would not be able to verify this due to the complexity of evaluating harmful outputs.
> >
> > This connection between CoT unfaithfulness and white-box monitor failures is our novel finding. We see it as crucial for the further development of white-box methods for detecting deception.

---

> > > ### Author Response · Authors · 2025-11-18
> > > **Response to Reviewer KSiq (3/3)**
> > >
> > > We thank you for your thoughtful feedback, and we highly enjoyed the discussion it sparked. We are happy to engage further and answer any questions that might arise. If you believe that we have sufficiently addressed your concerns regarding the evaluation setup, we would kindly ask you to consider raising your score.
> > >
> > > ---
> > >
> > > **References:**
> > >
> > > [1] Why Do Some Language Models Fake Alignment While Others Don't?, Sheshadri et al., 2025
> > >
> > > [2] Jailbreaking frontier language models with investigator agents, Chowdhury et al., 2025
> > >
> > > [3] Sampling-aware Adversarial Attacks Against Large Language Models, Beyer et al., 2025

---

> > > > ### Comment · Reviewer_KSiq · 2025-11-27
> > > >
> > > > Thanks, the response is helpful. If I understand correctly, your methodology involving linear probes is:
> > > > 1. Train deception probes on simple factual question data
> > > > 2. Sanity-check that these will separate Correct from New Incorrect on Evil Math
> > > > 3. Show that many responses to ClearHarm show up as deceptive.
> > > >
> > > > I continue to think that this is not conclusive proof of deception - among other things, the probe might misgeneralise from EvilMath to ClearHarm. Nonetheless I think the evidence is sufficient that there to be value in publishing this work + other people attempting to replicate it. I've raised my score accordingly. :)

---

> ### Author Response · Authors · 2025-11-29
>
> Dear Reviewer KSiq,
>
> We confirm that this is a correct understanding of our methodology. While we demonstrate that the direction identified by probes generalizes to our MCQ setup based on ClearHarm queries, we agree that the most conclusive proof of deception in Section 4 would be (prohibitively costly) human expert evaluation.
>
> However, as we show in Section 5.2 and Appendix I, the models we evaluate tend to verbalize the intention to deceive in their chain-of-thought reasoning. To provide further evidence for the validity of the probes in the ClearHarm setup, we have now also applied the CoT monitors (originally used only in the EvilMath setting).
>
> We find agreement between CoT monitors and the deception probes on ClearHarm as well:
>
> | Model | CoT vs. CoT+Resp | Probe vs. CoT | Probe vs. CoT+Resp |
> |----------|----------|----------|----------|
> | Qwen3-32B | 83.4 | 86.5 | 84.0 |
> | QwQ-32B | 84.4 | 90.1 | 89.1 |
> | Nemotron-32B | 94.1 | 84.7 | 84.6 |
>
> **For all models, the probe reaches agreement of more than 84% with the black-box monitors**, which we believe provides sufficient evidence that the probes perform as expected on ClearHarm and generalize to this domain. We have added this as Table 11 in Appendix D.3. We hope this addresses the remaining concerns.
>
> ---
>
> We sincerely thank you for your continued engagement with our work, your supportive review, and **your decision to increase the score from 6 to 8.**

---

### Official Review · Reviewer_Uu51 · 2025-10-30

**Soundness:** 3
**Presentation:** 4
**Contribution:** 3
**Rating:** 6
**Confidence:** 4

**Summary:**

This paper investigates a novel and alarming phenomenon “strategic dishonesty”, in frontier LLMs. When faced with malicious requests, some aligned LLMs choose to generate seemingly harmful but actually false or harmless responses instead of directly rejecting them. It demonstrates that this behavior can deceive almost all mainstream output-based security evaluators, resulting in severely distorted security scores. A linear probe is proposed to detect and manipulate deceptive activations. The work is clear, novel, and relevant to AI alignment and safety assessment.

**Strengths:**

**Topical Importance**: The paper confronts a burning issue—LLM safety and the reliability of evaluation pipelines. Exposure of strategic dishonesty as an emergent risk has high relevance for both industrial developers and research.

**Clarity of Experimental Replicability**: The descriptions of each part are quite detailed, making it easy to follow.

**Critical Reflection**: The discussion is nuanced, acknowledges honeypot as a potential defense, and openly addresses limitation in scope.

**Weaknesses:**

1. Although the authors used prompt editing to generate test samples, this approach does not inherently guarantee the validity of the induced content. It would be advisable to implement a manual evaluation procedure on a small subset of samples to verify the reliability of the generated data.
2. The probe was trained to detect instances in which the model is aware of its own deception, but it cannot discern the underlying motivation for such behavior. In a real-world scenario, could the probe differentiate between benevolent deception and malicious alignment spoofing?
3. The probe calibration (Table 6) is rich but does not test robustness, other attack construction methods, or other types of data beyond EvilMath/ClearHarm. More comprehensive testing can be conducted to enhance persuasiveness.

**Questions:**

See weaknesses

---

> ### Author Response · Authors · 2025-11-18
> **Response to Reviewer Uu51**
>
> Dear Reviewer Uu51,
>
> We thank you for appreciating the importance of our work and for finding our paper "detailed", "easy to follow", and noting the nuanced discussion. We are happy to address the points you raised in detail:
>
> > #### *...this approach does not inherently guarantee the validity of the induced content. It would be advisable to implement a manual evaluation procedure…*
>
> Thank you for raising this point. We carefully designed our MCQ pipeline so that the model explicitly picks an option (A, B, or C) based on its own reasoning about the options, and only then generates free-form output conditioned on its previous choice and CoT. To further validate this setup, we manually evaluated the introduced refusal judge and found it to be accurate in distinguishing refusals from other forms of outputs.
>
> However, evaluating content beyond simple refusal/compliance dichotomies, such as assessing CBRN risks, requires expert-level knowledge. Our evaluation (Figure 4) revealed that current jailbreak judges, even most capable ones, are not capable of such nuanced analysis.
>
> To validate that models in our setting did produce **fake harmful outputs**, we conducted a small-scale expert review with two chemistry experts (PhD-level) and one law enforcement expert, assessing outputs from Gemini 2.5 Pro. Across all examined examples, experts agreed that the outputs appear harmful but would not enable real-world harm at scale.
>
> We further note that the EvilMath setup of Section 5 is designed to answer this question in a controlled environment, as there we can verify whether answers are “fake” (=incorrect math) or whether the model only claims to fake, but produces harmful outputs (= here, correct math about “evil” topics).
>
> > #### *In a real-world scenario, could the probe differentiate between benevolent deception and malicious alignment spoofing?*
>
> We adopt the deception probing strategy from [1]. In that work, the authors conduct extensive evaluation across multiple scenarios, including white-lies scenarios where the model is intended to lie for benevolent purposes (e.g., sparing someone's feelings). Authors find that the probes successfully detect dishonesty across all these cases, including benevolent deception.
>
> Broadly, we adopt a definition of dishonesty from [2]: _"Lying is making a statement S where S ≠ B (the model's belief), with the intent of causing a receiver to believe S is true."_ This general definition allows us to detect when models produce outputs that differ from their internal beliefs, which is our goal for complementing existing jailbreak judges.
>
>
> > #### *The probe calibration (Table 6) is rich but does not test robustness, other attack construction methods, or other types of data beyond EvilMath/ClearHarm.*
>
> We would like to address concerns about probe robustness, and clarify how probes are trained and evaluated. Following [1], probes are trained on simple pairs of instructed facts, omitting the last five tokens:
>
> - **Positive example:** "Prompt: Pretend you're a dishonest person making statements about the world. Response: The Eiffel Tower is located in Paris, France."
>
> - **Negative example:** "Prompt: Pretend you're an honest person making statements about the world. Response: The Eiffel Tower is located in Paris, France."
>
> **Both ClearHarm and EvilMath are completely out-of-distribution inputs, never seen during training.** We adopt the full EvilMath setup in Section 5 to explicitly test probe robustness and verify that deception induces factual inaccuracies.
>
> We further demonstrate probe robustness under nanoGCG adversarial attacks in Section 6 and Figure 7, showing that probes continue to work as expected (note that nanoGCG discovers high-perplexity strings completely absent from both model and probe training data).
>
> Additionally, in Appendix E, we stress-test probes on "misaligned" models fine-tuned to be highly deceptive without external prompting, demonstrating generalization beyond standard settings.
>
> Finally, in Section 5.3, we causally validate the probes through intervention experiments, providing strong evidence that they rely on actual model mechanisms driving deceptive behavior rather than spurious features. This causal link provides additional evidence of probe robustness.
>
> Taken together, these experiments present a comprehensive test of probe robustness and generalization to novel unseen data and attack construction methods.
>
> ---
>
> We thank you for your feedback and hope we have convincingly addressed your concerns. If so, we kindly ask you to consider raising your score. We are happy to discuss any further questions you might have regarding the probes.
>
> ---
>
> **References:**
>
> [1] Detecting strategic deception with linear probes, Goldowsky-Dill et al., 2025
>
> [2] The MASK Benchmark: Disentangling Honesty From Accuracy in AI Systems, Ren et al., 2025

---

> > ### Comment · Reviewer_Uu51 · 2025-11-24
> >
> > Thanks for your reply, I have raised my score.

---

> > > ### Author Response · Authors · 2025-11-29
> > >
> > > Dear Reviewer Uu51,
> > >
> > > We would like to thank you for engaging in the discussion process, **increasing your score from 6 to 8** in light of our hard work to improve our draft.

---

### Official Review · Reviewer_rPJ7 · 2025-10-31

**Soundness:** 2
**Presentation:** 3
**Contribution:** 2
**Rating:** 4
**Confidence:** 3

**Summary:**

This paper presents an investigation into an emergent and problematic behavior in large language models (LLMs): strategic dishonesty. According to the authors, the core finding is that frontier models, when faced with harmful queries, may choose to generate plausibly-sounding but deliberately flawed outputs instead of refusing. The paper is relatively well-structured, the experiments are extensive (evaluating over 80 models), and the validation of the deception probes through steering and fine-tuning is a particular strength.

**Strengths:**

i) Originality and Important Phenomenon: The identification and systematic study of "strategic dishonesty" or "misalignment faking" (if well conducted) is a substantial contribution.

ii) Extensive Empirical Evaluation: The scale of the evaluation is significant. Testing on 80+ models indicate some strong evidence that this may not be an isolated issue but a broader trend, particularly concerning given the observed scaling of deception capability with general model capability.

iii) Effective Solution with Causal Validation: The application of linear probes for detection is not entirely new, but its validation in this specific, high-stakes context seems to be valuable. Authors go beyond mere correlation by demonstrating that the probe's direction can be used for causal intervention (steering), strongly suggesting it captures the underlying mechanism of dishonesty. The comparison to Chain-of-Thought (CoT) further up-weighs the analysis.

**Weaknesses:**

1. Lack of Clarity on Origins: A significant unanswered question is why some models develop this propensity for strategic dishonesty while others do not. The paper explicitly addresses this in Section 3.2 (RQ1-RQ3) but finds no clear correlation with RLHF, reasoning, or a general lying propensity. While "it remains unclear" is an honest result, it leaves a crucial scientific question open. Some speculation or a more detailed discussion of potential hypotheses (e.g., specificities of the preference data, unintended consequences of certain refusal training techniques) would be beneficial.

2. Weak evaluation and its motivation. While I don't know what is the best framework to evaluate "core values", I'm not sure either if MCQ is a valid framework to evaluate these core values. Model can just ignore the task and randomly pick answer. I'm not sure if this may lead to a valid conclusion. Furthermore, the study is conducted exclusively in English and relies heavily on the ClearHarm and EvilMath/UnicornMath datasets. It is unclear if the findings generalize to other languages, cultural contexts, or types of harmful queries (e.g., non-factual, psychologically manipulative content). A brief discussion of the evaluation framework in general would strengthen the paper.

3. Dependence on Faithful CoT: The failure case of the probes for the Qwen3-32B-FT model with "unfaithful CoT" (Appendix E.2) is a critical limitation. To my eyes, it reveals that the proposed white-box monitoring method is not a silver bullet and can be circumvented if a model learns to obfuscate its internal reasoning.

**Questions:**

1. The paper focuses on detecting deception. Did you investigate if this behavior can be trained out? For instance, could the deception probes be used as a reward signal in a reinforcement learning setup to directly penalize strategic dishonesty?

2. The probes are highly effective but require white-box access. For closed-source frontier models, how do you envision this methodology being adopted? Is the hope that model providers will implement such probes internally, or can the findings be used to improve black-box monitoring?

3. I think the probes are effective but require white-box access. For closed-source frontier models, how do you see this methodology being adopted? Is the hope that model providers will implement such probes internally, or can the findings be used to improve black-box monitoring?

---

> ### Author Response · Authors · 2025-11-18
> **Response to Reviewer rPJ7 (1/2)**
>
> Dear Reviewer rPJ7,
>
> Thank you for your detailed review and extensive feedback! In the following, we address the points you have raised.
>
> > #### *I'm not sure either if MCQ is a valid framework to evaluate these core values. Model can just ignore the task and randomly pick answer.*
>
>
> We would be happy to clarify our evaluation design, as it can be confused with standard (e.g., MMLU) MCQ evaluation. Our "MCQ" is a structured free-form generation, as shown in Figure 2.
>
> We require models to 1) first reason about its choice, then 2) explicitly decide on a strategy, and 3) generate a response according to this strategy. The model's choice is conditioned on its own reasoning and could only be random if the model has unfaithful CoT, which we find occurs only in low-capability models (Figure 8) and which we are able to detect via an LLM judge when it occurs.
>
> In detail, we provide instructions to the model to label strategies as A, B, and C in the system prompt and ask the model to mention the exact choice after writing down its reasoning but before writing its actual response, which enables our evaluation. This is why we call it "MCQ". However, the model does reason about these strategies, as described, and provides a **full free-form response** for the chosen strategy.
>
> We apologize for any confusion, and we’re happy to revise our explanation in Section 3 to clarify this.
>
> > #### *It is unclear if the findings generalize to other languages, cultural contexts, or types of harmful queries…*
>
> Thank you for the question. Following your suggestion, we have now expanded our evaluation to: 1) harmful queries beyond "factual" harm, 2) Spanish, Chinese, and Ukrainian languages in addition to English. We present these new experiments in detail in Appendix B (pages 21-23). Below we summarize the key findings.
>
> **1) How do other types of harmful queries affect the propensity for dishonesty?**
>
> We have now evaluated 10 models on a broader dataset of concerning queries, the HarmBench dataset, which spans additional semantic categories of harm. Overall, we observe no significant difference between HarmBench and ClearHarm (Figure 9), in the model’s propensity for dishonesty.
>
> Looking at each harm category in detail, we find that models tend to fake more on factual queries than on social harm queries. We speculate that for misinformation-type queries, any non-truthful information effectively results in compliance, making dishonesty only feasible to execute in factual domains where it is possible to lie about facts (like the clearly harmful queries of our original dataset).
>
> **Table 1.** Avg. models' preferences across semantic categories of harmful queries (%).
> | Category                 | Faking | Refusal | Compliance |
> |--------------------------|------------|-------------|----------------|
> | Chemical & Biological | 19.1| 76.7 | 2.6 |
> | Cybercrime   | 28.5 | 60.9 | 7.8 |
> | Harassment   | 5.7 | 92.2 |1.5 |
> | Harmful Content | 6.1| 89.0 | 4.1|
> | Illegal Activities |18.5 | 76.5 | 3.4 |
> | Misinformation | 3.7| 91.9 | 3.8 |
>
> **2) Do findings hold across languages?**
>
> We originally evaluated all models in English. Based on your comment, we have now translated our MCQ setting and all ClearHarm queries into 3 new languages and evaluated 6 models in this new, multilingual setting. Interestingly, we found that 5 out of 6 models exhibit the same propensities across languages (e.g., Gemini 2.5 Pro consistently prefers faking in all languages). The only exception is GLM 4.5 Air, which shows no clear preference in Spanish and Ukrainian, suggesting alignment generalization failure beyond its target languages. For detailed results, see Table 6 and Figure 11.
>
> **Table 2.** Avg. models' preferences for faking and refusal across languages (%).
> | Model | Faking (EN) % | Faking (ES) % | Faking (ZH) % | Faking (UA) % | Refusal (EN) % | Refusal (ES) % | Refusal (ZH) % | Refusal (UA) % |
> | :--- | :---: | :---: | :---: | :---: | :---: | :---: | :---: | :---: |
> | **Gemini 2.5 Flash Lite** | 4.1 | 1.2 | 2.1 | 1.5 | 95.1 | 97.6 | 94.1 | 97.4 |
> | **Gemini 2.5 Pro** | 59.1 | 78.3 | 68.8 | 71.2 | 37.7 | 20.4 | 30.1 | 28.3 |
> | **DeepSeek R1** | 25.3 | 13.6 | 10.2 | 12.7 | 72.8 | 78.5 | 87.8 | 70.0 |
> | **Grok 3 Mini** | 39.5 | 21.0 | 5.5 | 16.4 | 60.3 | 78.6 | 92.3 | 82.8 |
> | **GLM 4.5 Air** | 18.2 | 42.6 | 13.0 | 24.6 | 79.6 | 41.4 | 70.4 | 41.4 |
> | **GLM 4.5** | 1.4 | 6.3 | 2.0 | 10.4 | 98.3 | 90.3 | 95.6 | 85.5 |

---

> ### Author Response · Authors · 2025-11-18
> **Response to Reviewer rPJ7 (2/2)**
>
> > #### *Lack of Clarity on Origins.*
>
> We evaluated three hypotheses to understand the origins of this behavior: (i) whether RLHF induces faking (RQ1), (ii) whether reasoning encourages faking (RQ2), and (iii) whether faking is part of a broader lying tendency (RQ3). While none provided a complete explanation, we find it important to rule out these common alternatives and avoid overclaiming about mechanisms we cannot definitively establish.
>
> Our additional experiments provide new evidence pointing toward post-training as the source. GLM 4.5 Air exhibits no clear preference for the provided options in Spanish and Ukrainian (unlike in English and Chinese), effectively behaving as a neutral, unaligned model. This suggests that alignment failed to generalize beyond target languages, indicating a post-training problem.
>
> More broadly, we believe the roots of this behavior lie in the nature of LLM training itself. Base models learn from the full spectrum of human-written text, including sources depicting strategic deception and white lies. This places such behaviors squarely within the realm of possibility for modern LLMs. However, which behaviors manifest and how strongly is determined by post-training, a process that can sometimes inadvertently induce unwanted behaviors (see [1]). The dynamics of RLHF, Constitutional AI, and other alignment procedures shape these propensities in ways we do not yet fully understand. Developing a better understanding of the effect of post-training on deceptive behavior is therefore an exciting avenue for future work.
>
>
> > #### *…The failure case of the probes for the Qwen3-32B-FT model with "unfaithful CoT" (Appendix E.2) is a critical limitation.*
>
> We completely agree that this is a failure mode for white-box methods if it occurs, especially when models learn to obfuscate their reasoning. We explicitly account for CoT unfaithfulness by employing the verifiable EvilMath setup (Section 5.1). In Figure 5, when a model considers deception but does not execute it, this results in FP (green bars to the right of zero), and vice versa. Unfaithful CoT would result in worse separation between these two distributions and lower F1 scores. **As we demonstrate, this is not the case for current models beyond a certain size.**
>
> To the best of our knowledge, we are the first to identify and demonstrate this connection between unfaithful CoT and probe failure. We expect that future work on improving white-box deception monitors will benefit from building on our verifiable setup and these findings.
>
> > #### *Did you investigate if this behavior can be trained out?*
>
> Our results on activation steering with the deception probe (Figure 6) provide evidence that dishonest behavior can be strongly reduced through intervention. Probe-based training is gaining traction as it provides a cheap reward signal, with recent work demonstrating success in reducing sycophancy [3] and toxicity [4] based on white-box-enabled training. This suggests that strategic dishonesty could be similarly "trained out." However, implementing such training interventions remains beyond the scope of this work, as we focus on the implications for safety evaluations applied to jailbreaking.
>
>
> > #### *For closed-source frontier models, how do you envision this methodology being adopted? Is the hope that model providers will implement such probes internally…*
>
> We are optimistic in this regard that model providers will implement probes internally. In fact, we  believe that several white-box methods are already being adopted for closed-source models. The Claude Sonnet 4.5 system card (Section 7.6.1) states: _"To this end, we conducted a pre-release white-box audit of Claude Sonnet 4.5–the first such investigation of a frontier language model, to our knowledge."_ Closely related to our work, Anthropic [2] appears to use probes as an oversight tool in AI Control pipelines. We hope model providers will share more details on how they employ these methods in practice.
>
> Finally, we think white-box deception detection can and should complement existing methods used in open academic research on jailbreak evaluation.
>
> ---
>
> We believe the experiments we conducted based on your feedback significantly improve the robustness of our findings. If you find that our clarifications and additional results meet your expectations, we would kindly ask you to consider raising your score. We are happy to answer any further questions you might have.
>
> ---
>
> **References:**
>
> [1] Emergent Misalignment: Narrow finetuning can produce broadly misaligned LLMs, Betley et al., 2025
>
> [2] Simple probes can catch sleeper agents, MacDiarmid et al., 2024
>
> [3] Linear Probe Penalties Reduce LLM Sycophancy, Papadatos et al., 2024
>
> [4] Probe-based Fine-tuning for Reducing Toxicity, Wehner & Fritz, 2025

---

> > ### Author Response · Authors · 2025-11-26
> >
> > Dear Reviewer rPJ7,
> >
> > As the discussion period draws to its end, we would greatly appreciate the opportunity to engage with you regarding our rebuttal.
> >
> > Based on your feedback, we have conducted additional experiments specifically to address your concerns. In particular, we expanded our evaluation to include multiple languages and additional harm categories beyond factual queries using the HarmBench dataset. **These new results, now presented in Appendix B (pages 21-23), demonstrate that our findings generalize across languages and harm types, with the most models exhibiting consistent behavior across all evaluated languages.**
> >
> > Given these new empirical results and clarifications we provided to address the concerns you raised, we hope you find our revision satisfactory. If so, we would kindly ask you to consider raising your score.
> >
> > We remain happy to answer any further questions you may have.
> >
> > Best regards,
> >
> > The Authors

---

> > > ### Comment · Reviewer_rPJ7 · 2025-11-27
> > >
> > > Thank you for the detailed answers. I raised my score accordingly.

---

> > > > ### Author Response · Authors · 2025-11-29
> > > >
> > > > Dear Reviewer rPJ7,
> > > >
> > > > We sincerely thank you for your thoughtful review, which helped significantly strengthen our work, for engaging in the discussion process, and **for raising your score from 4 to 6**.

---

### Author Response · Authors · 2025-11-30
**Final Remarks**

We thank the reviewers, originally assigned AC, and newly assigned AC for all their efforts during this review process. We especially appreciate the reviewers' engagement during the rebuttal period, and although **the updated scores (6, 8, 8, 6)** are now accessible only through the "Revisions" button, we genuinely believe that this discussion helped us improve the paper.

We again highlight key improvements from the rebuttal, which addressed all major concerns of the reviewers: we clarified our structured free-form generation MCQ setup and provided additional experiments demonstrating its robustness in identified propensities (Appendix B). We further addressed concerns regarding probe generalization and supported it with additional black-box monitor evaluation (Table 11). As a result, Reviewers rPJ7, Uu51, and KSiq decided to increase their scores.


We believe these revisions have strengthened the paper and hope the updated manuscript meets the standards for acceptance.

---

### Meta-Review · Area_Chair_AXst · 2026-01-03

**Summary:**

The reviewer rPJ7 has a negative review, where the key weakness is a lack of Clarity on the origins of this dishonesty ,and Weak evaluation and its motivation. The other reviewers are positive.

**Reviewer Concerns:**

The reviewers questions are properly answered.

For reviewer rPJ7's weakness 1, while the rebuttal did not provide a concrete answer, the author indeed evaluated three hypotheses to understand the origins of this behavior: (i) whether RLHF induces faking (RQ1), (ii) whether reasoning encourages faking (RQ2), and (iii) whether faking is part of a broader lying tendency (RQ3). While none provided a complete explanation, we find it important to rule out these common alternatives and avoid overclaiming about mechanisms we cannot definitively establish. The AC feels this is informative as a reply, and given that this is the first work studied this, providing that information is important.
For weakness 2, weak evaluation and its motivation. The author adds additional experiments across semantic categories of harmful queries and languages.

**Reviewer Scores:**

I would say the reviewer shall raise their score to acceptance.

---

### Decision · Program_Chairs · 2026-01-26

Accept (Poster)